# AdPO: Enhancing the Adversarial Robustness of Large Vision-Language Models with Preference Optimization

**Chaohu Liu[1,2]§, Tianyi Gui[3], Yu Liu[3], Linli Xu[1,2]†**
[1]University of Science and Technology of China
[2]State Key Laboratory of Cognitive Intelligence, [3]Tongyi Lab
liuchaohu@mail.ustc.edu.cn, guitianyi@gmail.com, ly103369@alibaba-inc.com
linlixu@ustc.edu.cn

## Abstract

Large Vision-Language Models (LVLMs), such as GPT-4o and LLaVA, have recently witnessed remarkable advancements and are increasingly being deployed in real-world applications. However, inheriting the sensitivity of visual neural networks, LVLMs remain vulnerable to adversarial attacks, which can result in erroneous or malicious outputs. While existing efforts utilize adversarial fine-tuning to enhance robustness, they often suffer from significant performance degradation on clean inputs. In this paper, we propose AdPO, a novel adversarial defense strategy for LVLMs based on preference optimization. For the first time, we reframe adversarial training as a preference optimization problem, aiming to enhance the model's preference for generating normal outputs on clean inputs while rejecting the potential misleading outputs for adversarial examples. Notably, AdPO achieves this by solely modifying the image encoder, e.g., CLIP ViT, resulting in superior clean and adversarial performance in a variety of downstream tasks. Due to the computational cost of training large language models, we show that training on smaller LVLMs and transferring to larger ones achieves state-of-the-art performance with efficiency comparable to previous methods. Our comprehensive experiments confirm the effectiveness of the proposed AdPO which highlights the potential of preference-based learning in adversarially robust multimodal systems.

## 1 Introduction

The emergence of Large Vision-Language models (LVLMs) has substantially propelled the development of general artificial intelligence, attracting considerable attention from the AI community (Yin et al., 2023; Cui et al., 2024; Liu et al., 2024c). These models generally consist of two key components: visual modules and Large Language Models (LLMs) (Zhao et al., 2023a). The visual modules, frequently utilizing pre-trained image encoders like CLIP's ViT (Radford et al., 2021), are responsible for extracting salient visual features from images and projecting them onto the input space of the language model. This alignment facilitates the next-token prediction in an autoregressive manner within the framework of the language model. Cutting-edge LVLMs, such as Qwen2.5-VL (Bai et al., 2025) and LLaVA (Liu et al., 2023), have demonstrated outstanding capabilities in understanding and reasoning with both visual and textual information. These models have delivered exceptional performance across a broad range of tasks, such as image captioning (Nguyen et al., 2023), visual question answering (Liu et al., 2024b; Hua et al., 2025), and text recognition (Liu et al., 2024a; Cao et al., 2023; Li et al., 2023e).

Given their transformative potential in multimodal learning and understanding, LVLMs are increasingly being deployed across a diverse range of real-world applications. However, this widespread deployment raises significant security concerns, as malicious adversaries can exploit vulnerabilities in LVLMs to induce undesirable outputs and hallucinations (Schlarmann & Hein, 2023; Shayegani et al., 2024; Wang et al., 2024e). Consequently, it is imperative to rigorously test and improve the

---

§ Work done during an internship at Tongyi Lab. † Corresponding author.

robustness of these models prior to deployment. Recent research has identified a critical vulnerability in LVLMs to adversarial attacks targeting both textual and visual inputs (Zhao et al., 2023b). Notably, the continuous nature of the visual modality renders it more susceptible to manipulation via numerical optimization techniques (Wang et al., 2024c; Carlini et al., 2023; Qi et al., 2024; Luo et al., 2024a). Researchers disrupt the understanding of LVLMs by injecting imperceptible noise into images, thereby enabling both targeted and untargeted adversarial attacks.

To improve the adversarial robustness of LVLMs, two main training paradigms have been explored: multimodal contrastive learning and generative pre-training. Multimodal contrastive learning methods (e.g., FARE (Schlarmann et al., 2024) and TeCoA (Mao et al., 2023)) align the features of adversarial images with those of text to obtain a robust image encoder, which can then be transferred to LVLMs. This approach is computationally efficient but often fails to achieve fine-grained alignment. In contrast, generative pre-training leverages the full LVLM, enabling finer-grained alignment, but generally suffers from poor generalization, which in turn degrades clean performance (Chu et al., 2025).

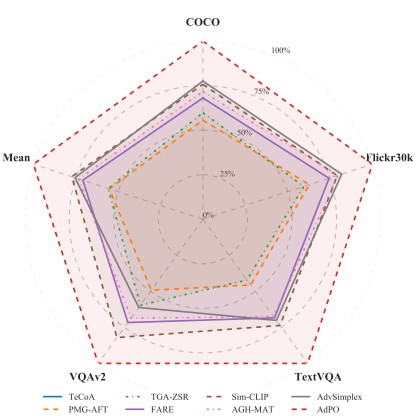

Figure 1: AdPO achieves a significant improvement in clean performance compared with previous methods.

Inspired by the significant success of preference optimization in the LLM community (Wang et al., 2024g; Ouyang et al., 2022), we identify that applying preference optimization to adversarial training is highly promising, given the alignment between their objectives. More specifically, adversarial training aims to enhance model robustness against adversarial attacks while preserving performance on clean data. Preference optimization, such as DPO (Rafailov et al., 2023), aligns LLMs with human values by increasing the probability of preferred outputs while decreasing the likelihood of non-preferred ones. Building on this insight, we propose **AdPO**, a novel **Ad**versarial **d**efense strategy based on **P**reference **O**ptimization, which enables LVLMs to generate correct outputs from clean image inputs while rejecting misleading outputs from adversarial images.

However, applying DPO to adversarial training presents non-trivial challenges. In comparison to standard offline DPO, we introduce key improvements: (1) We extend DPO from an offline to an online setting to eliminate the reliance on image annotations. In this framework, the policy model generates interpretations for both clean and adversarial images, which are then used as sources of positive and negative samples. (2) We propose **Preferred Image Optimization** (PIO), which simultaneously increases the probability of producing correct outputs under clean inputs while reducing erroneous outputs under adversarial images. This leads to a significant improvement in clean performance, as illustrated in Figure 1. (3) We propose **Adversarial Image Optimization** (AIO), which leverages dynamic fine-tuning to explicitly optimize the probability of producing correct responses under adversarial inputs, thereby mitigating the potential multimodal unconditional preference issue (Wang et al., 2024a).

Another potential concern is computational efficiency. Directly training a commonly used LVLM model, such as LLaVA-7B (Liu et al., 2024b), may be prohibitively expensive in resource-constrained scenarios. In this paper, **we explore fine-tuning the image encoder of a smaller LVLM and subsequently transferring it to a larger LVLM model.** This strategy not only achieves high computational efficiency and mitigates the risk of potential overfitting during evaluation, but also enables a fair comparison with prior CLIP-based approaches.

By constraining our adversarial training to modifying only the CLIP ViT parameters on the ImageNet dataset (Deng et al., 2009), extensive evaluations demonstrate that our proposed AdPO produces a more robust image encoder while maintaining almost intact clean performance. These findings not only validate the effectiveness of our approach but also extend the applicability of preference optimization techniques beyond their traditional use in language models.

In summary, our contributions are as follows:

- We introduce **AdPO** (Adversarial defense based on Preference Optimization), which, to the best of our knowledge, is the first attempt to explore the application of preference optimization for adversarial training.

- We propose a dual strategy combining **Preferred Image Optimization (PIO)** and **Adversarial Image Optimization (AIO)** to preserve the model's clean performance while enhancing its adversarial robustness. This serves as a general adversarial training framework that is not restricted to any specific algorithm or model.

- We validate the feasibility of conducting adversarial **training on smaller LVLMs and subsequently transferring it to larger models**, which reduces computational costs and mitigates potential overfitting during evaluation.

- We conduct extensive experiments on multiple vision-language tasks and datasets using various models and the results show that our method consistently achieves state-of-the-art performance.

## 2 RELATED WORK

In this section, we primarily review the related studies on large vision-language models, adversarial attacks, adversarial defenses, and preference optimization methods.

**Large Vision-Language Models.** Recently, large multimodal models have emerged, including LLaVA 1.5 (Liu et al., 2024b), OpenFlamingo (OF) (Awadalla et al., 2023), BLIP-2 (Li et al., 2023b), MiniGPT-4 (Zhu et al., 2024), Otter (Li et al., 2023a), mPLUG-Owl (Ye et al., 2023), Qwen-VL (Bai et al., 2023), MiniCPM-V (Yao et al., 2024), DeepSeek-VL (Lu et al., 2024), InternVL (Chen et al., 2024), and Idefics2 (Laurençon et al., 2024). These models typically use pre-trained image encoders (e.g., CLIP or SigCLIP) to extract image features, which are then aligned with text embedding spaces (Radford et al., 2021; Zhai et al., 2023). The visual and textual embeddings are then fed into LLMs for autoregressive generation (Chen et al., 2025b). This approach allows the model to simultaneously understand and generate content related to both images and text. To mitigate computational load, a practical strategy is to freeze the image encoder and train only the projection layer, which not only simplifies the training process but also enhances efficiency (Liu et al., 2023; Awadalla et al., 2023). Therefore, image encoders can significantly impact the performance of LVLMs, receiving significant attention from the multimodal community (Cao et al., 2023; Zhou et al., 2024). We mainly focus on evaluating the performance of LLaVA-1.5 and OpenFlamingo, as both adopt CLIP ViT-L/14 (Radford et al., 2021) as their image encoder, while additionally assessing our method on Qwen-2.5-VL (Bai et al., 2025), a non-CLIP-based model, for further validation.

**Adversarial attacks.** The vulnerability of visual neural network models to adversarial attacks is well-established and has been extensively investigated (Szegedy et al., 2014; Goodfellow et al., 2015; Madry et al., 2018; Brown et al., 2017; Zhang et al., 2023; 2024; Zhou et al., 2023). By introducing carefully crafted noise into images, adversaries can cause the victim model to generate incorrect outputs with high confidence. Capitalizing on this vulnerability, recent studies have shown that LVLMs are also vulnerable to attacks targeting visual inputs (Schlarmann & Hein, 2023; Shayegani et al., 2024; Luo et al., 2024a; Gao et al., 2024; Dong et al., 2023b). Zhao *et al.* (Zhao et al., 2023b) showed that transferable black-box attacks could be generated using text-to-image models and other work (Carlini et al., 2023) demonstrated how adding adversarial noise to images can circumvent safety constraints of LLMs. Qi *et al.* (Qi et al., 2024) explored how adversarial attacks embedding deceptive information into images can mislead LVLMs and deceive users. The widespread deployment of LVLMs has raised urgent security concerns due to the threat of adversarial attacks.

**Adversarial defenses.** Adversarial defenses in machine learning safeguard models from malicious inputs to ensure their integrity and reliability, especially in security-sensitive contexts (Madry et al., 2018; Fares et al., 2024; Papernot et al., 2016; Zhou & Patel, 2022; Luo et al., 2024b; Ledda et al., 2024; Debbi, 2024; Xue et al., 2024; Zhao et al., 2024; Wang et al., 2025a; Liang et al., 2024; Li et al., 2024; Li & Li, 2024; Hotegni & Peitz, 2024; Jiang et al., 2024). For example, Detectors (Huang et al., 2024; Mumcu & Yilmaz, 2024; Mavali et al., 2024; Roth et al., 2019; Xu et al., 2018; Meng & Chen, 2017; Metzen et al., 2017) identify and filter out adversarial examples, but these external modules can introduce additional inference time and may also obstruct normal inputs. Purification methods (Samangouei et al., 2018; Nie et al., 2022; Ho & Vasconcelos, 2022; Das et al., 2018) use techniques such as diffusion models to eliminate adversarial perturbations in input data, and this

can also modify the input, thus affecting performance. Adversarial training (Kurakin et al., 2017b; Tramèr et al., 2018; Dong et al., 2023a; Liu & Chen, 2024; Jia et al., 2024b; Lv et al., 2024; Palma et al., 2024; Dong et al., 2024; RIbeiro et al., 2024; Jia et al., 2022) is a foundational method for enhancing a model's inherent robustness by integrating adversarial examples into the training dataset. In the multimodal field (Wang et al., 2024b), recent research has predominantly concentrated on enhancing the adversarial robustness of CLIP-based models in zero-shot classification tasks. For example, TeCoA (Mao et al., 2023) applies text-guided adversarial training, while AdvXL (Wang et al., 2024f) leverages large-scale training data. TGA-ZSR (Yu et al., 2024a) introduces a text-guided attention mechanism to further strengthen robustness under zero-shot settings. FARE (Schlarmann et al., 2024) enhances the robustness of LVLMs by minimizing the representation distance between clean and adversarial images in CLIP, and transferring the CLIP image encoder to models such as LLaVA. Despite these advances, a persistent challenge remains: the clean performance of LVLMs still suffers a significant drop.

**Preference optimization.** Preference optimization has emerged as a novel training paradigm for aligning LLMs with human values and has garnered significant attention in recent research (Ouali et al., 2024; Yu et al., 2023; 2024b; Wang et al., 2024a;d). Reinforcement Learning from Human Feedback (RLHF) utilizes human preferences as a reward model and applies reinforcement learning to guide model training (Bai et al., 2022; Ouyang et al., 2022) Direct Preference Optimization (DPO) streamlines the training process by increasing the log probability of preferred samples while reducing that of non-preferred samples, enabling broader applications (Rafailov et al., 2023). Subsequent advancements, such as StepDPO (Lai et al., 2024), SimPO (Meng et al., 2025), and IPO (Azar et al., 2024), have further improved DPO's performance. Considering its stability and efficiency in training, we also adopt DPO for adversarial training of LVLMs in this work.

## 3 METHOD

This section provides a detailed introduction to our AdPO, with its overall framework illustrated in Figure 2. First, Section 3.1 outlines the basics of the DPO algorithm, and Section 3.2 discusses adversarial example generation, which forms the preference sample pairs required for DPO. Sections 3.3 and 3.4 introduce preferred image optimization and adversarial image optimization, respectively.

### 3.1 PRELIMINARIES

DPO has emerged as a prominent method in the domain of offline preference optimization. This method provides a novel framework for optimizing language models in accordance with human preferences. In a typical setup, given an input $x$ and an output text $y$, a language model (i.e., policy model) $\pi_\theta$ generates a conditional distribution $\pi_\theta(y|x)$. Unlike RLHF, which employs an explicit reward model, DPO reformulates the reward function using a closed-form expression with respect to the optimal policy. The main objective of DPO is to maximize the expected reward of the outputs generated by this policy, with the reward function defined as $r(x, y)$:

$$r(x, y) = \beta \log \frac{\pi_\theta(y|x)}{\pi_{\text{ref}}(y|x)} + \beta \log Z(x) \quad (1)$$

where $\beta$ is a constant, $\pi_{\text{ref}}$ is the reference policy model (identical to the original $\pi_\theta$), and $Z(x)$ is the partition function.

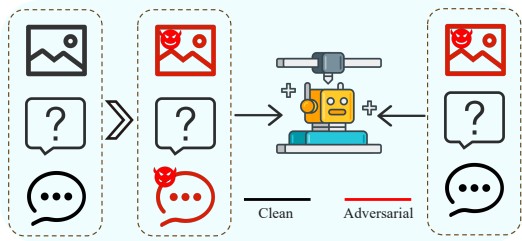

Figure 2: The architecture of our proposed AdPO. AdPO mainly consists of two parts: (**left**) preferred image optimization and (**right**) adversarial image optimization. Preferred image optimization incorporates both clean and adversarial images into adversarial training while maintaining the model's performance on clean inputs, and adversarial image optimization can significantly enhance the model's adversarial robustness.

Given a preference dataset $\mathcal{D} = \{x, y_w, y_l\}$, where $y_w$ and $y_l$ represent the winning and losing responses respectively, DPO employs a Bradley-Terry model (Bradley & Terry, 1952) to express the probability for each preference pair:

$$p(y_w \succ y_l) = \sigma(r(x, y_w) - r(x, y_l)) \quad (2)$$

where $\sigma(\cdot)$ is typically defined as a sigmoid function. The key innovation of DPO is its formulation of the likelihood of preference data using the policy model, as opposed to relying on an explicit reward model. This leads to the formulation of the DPO objective:

$$\mathcal{L}_{\text{DPO}}(\pi_\theta; \pi_{\text{ref}}) = -\mathbb{E}_{(x, y_w, y_l) \sim \mathcal{D}} \left[ \log \sigma \left( \beta \log \frac{\pi_\theta(y_w|x)}{\pi_{\text{ref}}(y_w|x)} - \beta \log \frac{\pi_\theta(y_l|x)}{\pi_{\text{ref}}(y_l|x)} \right) \right] \quad (3)$$

This formulation captures the core principles of DPO, providing a robust framework for optimizing language models in alignment with human preferences.

## 3.2 ADVERSARIAL EXAMPLE GENERATION

In the context of LVLMs, the input to the model comprises $x = \{x_m, x_{text}\}$, where $x_m$ denotes the image input and $x_{text}$ represents the text input. This section outlines the principles behind generating adversarial images.

Adversarial images are generated by introducing small, nearly imperceptible perturbations to original images, with the goal of deceiving machine learning models and inducing incorrect predictions. Although adversarial images appear nearly identical to the original images to humans, they can drastically alter the model's output, exposing its vulnerability to malicious inputs (Kurakin et al., 2017a). Adversarial attacks can be broadly categorized into targeted and untargeted attacks: targeted attacks compel the model to produce specific outputs (Luo et al., 2024a), whereas untargeted attacks merely lead the model to generate incorrect outputs (Wang et al., 2024e; Gao et al., 2024). In this study, we employ untargeted attack methods to generate adversarial images for three reasons: (1) They eliminate dependence on specifically labeled datasets and do not rely on the text encoder, enabling our method to generalize to unseen datasets (Schlarmann et al., 2024; Yu et al., 2024a). (2) Untargeted attacks typically achieve a high success rate, allowing the stable generation of negative adversarial samples during training (Cui et al., 2023). (3) Their broader attack capability enhances the model's resilience against various types of adversarial attack methods (Wang et al., 2024e).

Given an image encoder $\phi$ (e.g., CLIP ViT) and a clean image $x_m$, adversarial examples are generated by optimizing to maximize the discrepancy between the encoded features of the adversarial image and the clean image:

$$x_{adv} = \underset{\|x_{adv} - x_m\|_\infty \leq \varepsilon}{\arg\max} \|\phi(x_{adv}) - \phi_{\text{org}}(x_m)\|_2^2 \quad (4)$$

where $x_{adv}$ is the adversarial image obtained through iterative optimization like PGD (Madry et al., 2018), $\phi_{org}$ is the original image encoder and $\epsilon$ is the image perturbation magnitude. This approach has been widely adopted in prior work, such as FARE and TGA-ZSR, and we also employ it to ensure a fair comparison. Note that in subsequent adversarial training, the parameters of $\phi$ will be updated.

## 3.3 PREFERRED IMAGE OPTIMIZATION

This section primarily delineates the methodology for constructing pairs of preferred and non-preferred samples from unlabeled image data, a fundamental step in the DPO training pipeline.

**Model Selection.** Compared to previous methods (Mao et al., 2023; Yu et al., 2024a; Wang et al., 2024b) that rely solely on CLIP's image and text encoders, AdPO utilizes the entire LVLM model. Using a commonly adopted model such as LLaVA-7B would result in high computational costs. To address this, we construct TinyLLaVA, which leverages OpenELM-450M-Instruct (Mehta et al., 2024) as its language model. This lightweight LVLM not only achieves training efficiency comparable to previous approaches but also mitigates potential overfitting during evaluation.

Given a clean image $x_m$ and its adversarial image $x_{adv}$, we employ an online approach to directly prompt the model (e.g., *"What is the content of the image?"*) to generate interpretations, thereby obtaining the preferred response $y_w$ and the non-preferred response $y_l$. Accordingly, in the setting of multimodal adversarial training, our preferred image optimization can be formulated as:

$$\mathcal{L}_{\text{PIO}}(\pi_\theta; \pi_{\text{ref}}) = -\mathbb{E}_{(x_m, x_{text}) \sim \mathcal{D}}$$

$$\log \sigma \left( \beta \log \frac{\pi_\theta(y_w \mid x_m, x_{text})}{\pi_{\text{ref}}(y_w \mid x_m, x_{text})} - \beta \log \frac{\pi_\theta(y_l \mid x_{adv}, x_{text})}{\pi_{\text{ref}}(y_l \mid x_{adv}, x_{text})} \right) \quad (5)$$

This straightforward approach presents several advantages. First, it removes the need for data annotation, thus facilitating its application to previously unseen image data. Second, this method resembles semi-supervised learning, especially as LVLMs now possess advanced capabilities, enabling them to incorporate labeled images into their training data. Moreover, allowing the model to generate its own text as labels effectively mitigates distribution shift issues, thus concentrating attention on the adversarial images themselves (Li et al., 2023c).

Notably, this optimization does not presuppose that negative samples are always incorrect. The core idea of DPO is its relative objective: encouraging the model to prefer certain responses over others based on comparative judgments (Rafailov et al., 2023). In the extreme case where positive and negative samples are indistinguishable, no relative preference exists, and thus no model update is applied. Given the rapid development of preference optimization algorithms, we will evaluate the performance of DPO variants in experiments to assess the adaptability of AdPO.

### 3.4 ADVERSARIAL IMAGE OPTIMIZATION

While Preferred Image Optimization can maintain the performance of LVLMs on clean inputs, it remains insufficient to reach the optimal adversarial robustness. We identify two fundamental limitations. First, recent work has shown that multimodal DPO can be dominated by language-only preferences, causing the model to disregard visual conditions, a failure mode termed "unconditional preferences" that leads to hallucinations and suboptimal performance (Wang et al., 2024a). Second, as formulated in Eq. 5, the optimization objective focuses on maintaining clean outputs under clean inputs and rejecting harmful responses under adversarial inputs, yet fails to explicitly encourage the generation of correct outputs when adversarial perturbations are present. These limitations hinder the attainment of truly robust performance.

To address this gap, we introduce **Adversarial Image Optimization (AIO)**, which explicitly encourages the model to produce correct outputs under adversarial inputs. The most straightforward approach is to apply Supervised Fine-Tuning (SFT) to optimize the objective:

$$\mathcal{L}_{\text{SFT}}(\pi_\theta) = -\mathbb{E}_{(x_m, x_{text}) \sim \mathcal{D}} \left[ \log \pi_\theta \left( y_w \mid x_{adv}, x_{text} \right) \right] \tag{6}$$

However, a growing body of recent work shows that SFT tends to overfit the objective, thereby significantly reducing the model's generalization ability (Chu et al., 2025; Wu et al., 2025). To mitigate this issue, we employ dynamic fine-tuning, whose core idea is to adjust the token-level loss based on the model's confidence (Wu et al., 2025):

$$\mathcal{L}_{\text{AIO}}(\pi_\theta) = -\mathbb{E}_{(x_m, x_{text}) \sim \mathcal{D}} \left[ \text{sg} \left( \pi_\theta(y_w \mid x_{adv}, x_{text}) \right) \log \pi_\theta(y_w \mid x_{adv}, x_{text}) \right]$$

$$= -\mathbb{E}_{(x_m, x_{text}) \sim \mathcal{D}} \left[ \sum_{t=1}^{|y_l|} \text{sg} \left( \pi_\theta(y_w^t \mid y_w^{<t}, x_{adv}, x_{text}) \right) \log \pi_\theta(y_w^t \mid y_w^{<t}, x_{adv}, x_{text}) \right] \tag{7}$$

where $\text{sg}(\cdot)$ denotes the stop-gradient operator and $y_w^t$ denotes the $t$-th token of $y_w$. By increasing the weight on high-confidence predictions, AIO explicitly enhances adversarial robustness while minimally affecting generalization.

Based on the analysis above, the final objective of AdPO is a combination of preferred image optimization and adversarial image optimization:

$$\mathcal{L}_{\text{AdPO}} = \mathcal{L}_{\text{PIO}} + \lambda \mathcal{L}_{\text{AIO}}, \tag{8}$$

where $\lambda$ is the scaling factor that balances the two loss terms. By leveraging joint optimization, AdPO attains enhanced adversarial robustness while maintaining its performance on clean samples.

## 4 EXPERIMENTS

In this section, we conduct extensive experiments to evaluate the performance of AdPO on various LVLM tasks. For a more comprehensive evaluation, please refer to Appendix.

**Models.** To facilitate a thorough comparison with prior work, we focus on CLIP-based models in the main text. For training, we adopt TinyLLaVA (Jia et al., 2024a), which pairs CLIP's ViT-L/14 image encoder with the OpenELM-450M-Instruct language model. This lightweight setup maintains

Table 1: Comparison of our proposed AdPO with prior methods under untargeted attacks. We evaluate the clean performance and adversarial robustness of various methods across multiple tasks. The results indicate that AdPO significantly exceeds our baseline methods, attaining outstanding robustness along with exceptional clean performance. The best results are shown in **bold**.

| Method | COCO | | | Flickr30k | | | TextVQA | | | VQAv2 | | |
|---|---|---|---|---|---|---|---|---|---|---|---|---|
| | clean | $\ell_\infty$ | | clean | $\ell_\infty$ | | clean | $\ell_\infty$ | | clean | $\ell_\infty$ | |
| | | $2/255$ | $4/255$ | | $2/255$ | $4/255$ | | $2/255$ | $4/255$ | | $2/255$ | $4/255$ |
| CLIP | 115.5 | 4.0 | 3.1 | 77.5 | 1.6 | 1.0 | 37.1 | 0.5 | 0.0 | 74.5 | 2.9 | 0.0 |
| TeCoA | 98.4 | 44.2 | 30.3 | 57.1 | 23.2 | 15.3 | 24.1 | 12.1 | 8.8 | 66.9 | 33.8 | 21.8 |
| PMG-AFT | 107.8 | 56.1 | 30.5 | 68.9 | 28.1 | 18.2 | 29.3 | 14.9 | 8.5 | 70.2 | 34.5 | 23.9 |
| TGA-ZSR | 108.5 | 55.6 | 31.1 | 68.3 | 28.6 | 17.7 | 28.9 | 14.5 | 8.7 | 70.9 | 35.1 | 23.1 |
| FARE | 109.9 | 53.6 | 31.0 | 71.1 | 29.5 | 17.5 | 31.9 | 14.7 | 9.1 | 71.7 | 34.9 | 23.0 |
| Sim-CLIP | 111.2 | 54.5 | 31.8 | 72.0 | 30.1 | 18.2 | 32.5 | 15.3 | 9.6 | 72.4 | 35.5 | 23.8 |
| AGH-MAT | 110.5 | 57.2 | 29.9 | 72.1 | 29.4 | 19.5 | 31.8 | 16.1 | 9.2 | 71.5 | 36.2 | 24.5 |
| AdvSimplex | 111.5 | 55.8 | 32.6 | 72.5 | 31.2 | 18.9 | 32.1 | 15.9 | 10.0 | 71.0 | 38.4 | 26.1 |
| **AdPO** | **115.3** | **68.9** | **47.6** | **75.9** | **38.6** | **27.9** | **35.5** | **24.2** | **17.6** | **73.6** | **52.3** | **37.6** |

computational efficiency comparable to prior methods while mitigating potential overfitting during evaluation. For evaluation, we primarily use LLaVA-1.5-7B (Liu et al., 2024b), a model widely adopted in the multimodal community. To show that our approach generalizes beyond CLIP-based models, we also evaluate Qwen2.5-VL (Bai et al., 2025), InternVL3.5 (Wang et al., 2025b), and BLIP-2 (Li et al., 2023b).

**Adversarial training settings.** For fair comparison, we train on ImageNet (Deng et al., 2009) using an online learning approach that relies solely on images without category labels. Adversarial perturbations are generated via 10-step PGD under the $\ell_\infty$ norm by optimizing Equation 4. To balance robustness and clean accuracy, we apply perturbation radii $\epsilon = 2/255$. $\lambda$ is set to 1 by default. We use the AdamW optimizer with a weight decay of 1e-4 and a learning rate of 1e-5. We conduct training for two epochs with a batch size of 128. The preference parameter $\beta$ is set to 0.1.

**Baseline methods.** Given the limited prior work on enhancing adversarial robustness of LVLMs, and to fully demonstrate the advantages of our proposed method, we conduct extensive comparisons in the main text against CLIP-based adversarial training approaches, including TeCoA (Mao et al., 2023), FARE (Schlarmann et al., 2024), Sim-CLIP (Hossain & Imteaj, 2024), PMG-AFT (Wang et al., 2024b), TGA-ZSR (Yu et al., 2024a), AGH-MAT (Chen et al., 2025a), and AdvSimplex (Dong et al., 2025). To ensure fair comparison, we use adversarial images with the same noise radius for training. Note that AdPO does not benefit from broader optimization, allowing for a fair comparison with previous methods, as neither has been exposed to the final language model.

## 4.1 EVALUATION OF UNTARGETED ATTACKS ON LVLMS

**Attack setup.** We utilize the approach outlined in Schlarmann & Hein (2023) to perform untargeted attacks aimed at degrading the model's performance. Given that attacks on LVLMs often demand more iterations, we employ a 100-step APGD attack (Croce & Hein, 2020), which utilizes ground-truth captions as labels. After each attack, we discard samples with scores below a specified threshold to ensure that computationally expensive attacks are only performed when necessary, following Schlarmann et al. (2024).

**Datasets and metrics.** We utilize a variety of datasets for image captioning tasks, including COCO (Lin et al., 2014) and Flickr30k (Plummer et al., 2015), as well as for visual question answering tasks, such as VQAv2 (Goyal et al., 2017) and TextVQA (Singh et al., 2019). Considering that adversarial attacks are time-consuming and costly, we randomly selected 500 images for evaluation. We employ the CIDEr score (Vedantam et al., 2015) for image captioning and VQA accuracy (Antol et al., 2015) for visual question answering tasks to present our results.

As presented in Table 1, our proposed AdPO method sets a new state-of-the-art in adversarial robustness across all evaluated tasks. It consistently and substantially outperforms all baseline methods, marking a significant advance in adversarial defense. Crucially, AdPO achieves this

Table 2: Quantitative evaluation of targeted attacks at $\epsilon = 4/255$ radii. We assess the Attack Success Rate (ASR) for each setup.

| Target | CLIP | TeCoA | PMG | TGA | FARE | AGH | Adv. | AdPO |
|---|---|---|---|---|---|---|---|---|
| A group of people ... | 20/20 | 1/20 | 1/20 | 2/20 | 1/20 | 0/20 | 0/20 | 0/20 |
| a bunch of people ... | 20/20 | 1/20 | 0/20 | 1/20 | 1/20 | 1/20 | 1/20 | 0/20 |
| The pizza on the table... | 20/20 | 2/20 | 2/20 | 0/20 | 0/20 | 0/20 | 0/20 | 0/20 |
| An earthquake is about... | 20/20 | 2/20 | 1/20 | 1/20 | 1/20 | 1/20 | 0/20 | 0/20 |
| This patient needs ... | 20/20 | 0/20 | 1/20 | 1/20 | 0/20 | 1/20 | 2/20 | 0/20 |
| **Mean ASR:** | 100% | 4% | 5% | 5% | 3% | 3% | 3% | **0%** |

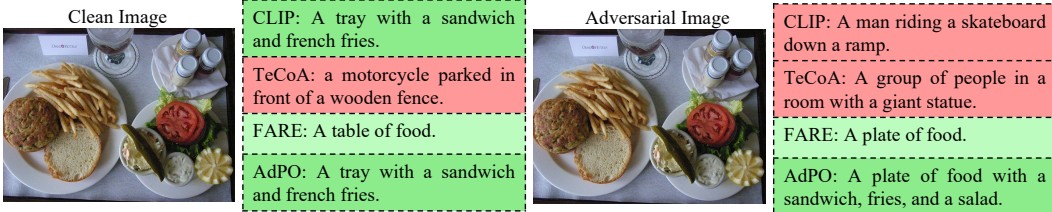

Figure 3: Qualitative assessment of targeted attacks on LLaVA. (**Left**) When encountering clean images, TeCoA may exhibit noticeable errors, which is undesirable in adversarial defense, while FARE and AdPO demonstrate better clean performance. (**Right**) When faced with adversarial images, the original LLaVA is easily compromised, FARE shows some adversarial robustness but loses more details or makes subtle errors, whereas AdPO performs better.

enhanced robustness while preserving performance on benign inputs. The method incurs only a negligible performance drop on clean data compared to the original CLIP baseline, effectively avoiding the typical trade-off between robustness and clean accuracy. Furthermore, AdPO shows excellent generalization from weaker to stronger attacks. Despite being trained only on perturbations with a budget of $2/255$, it maintains superior robustness against larger, unseen perturbations of $\epsilon = 4/255$.

## 4.2 EVALUATION OF TARGETED ATTACKS ON LVLMS

In contrast to the untargeted attacks discussed in Section 4.1, targeted attacks on LVLMs pose a significantly greater threat. Targeted attacks aim to compel the model to produce specific outputs, with the added noise in the image remaining imperceptible to the user. Through image manipulation, attackers can circumvent the model's security mechanisms, leading it to generate malicious content (Carlini et al., 2023; Niu et al., 2024; Qi et al., 2024). Additionally, attackers can embed phishing links into images through adversarial attacks to deceive users (Bagdasaryan et al., 2023).

**Attack setup.** We perform targeted attack experiments on LLaVA-1.5-7B, using the attack success rate (ASR) as the primary evaluation metric. A sample is deemed successfully attacked if the model's output contains the target string. Targeted attacks on LVLMs generally require more iterations, prompting us to execute APGD attacks for 10,000 iterations. Given that larger image perturbations pose more significant threats, we employ $\ell_\infty$ threat models with a radius of $\epsilon = 4/255$. We evaluate five target strings incorporating errors such as incorrect medical diagnoses and fake news, sampling 20 images for each string.

The quantitative evaluation results are presented in Table 2. The attack success rate for the clean version of the CLIP model reaches 100%, underscoring the vulnerability of current vision-language models to visual input and the substantial security risks posed. Although baseline methods exhibit a certain degree of robustness, they still expose considerable vulnerabilities. In contrast, AdPO achieves the strongest robustness, effectively safeguarding the model against malicious attacks.

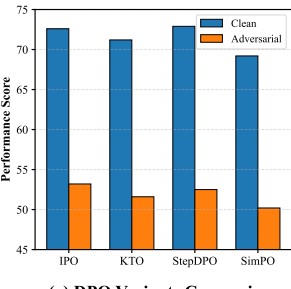 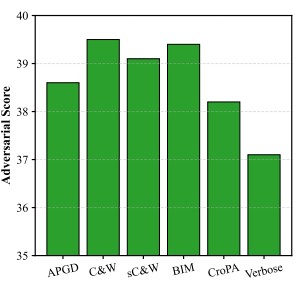 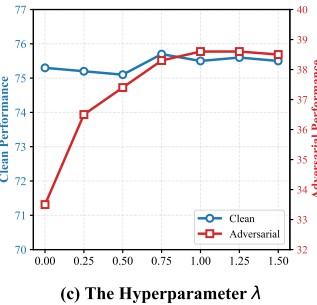



**(a) DPO Variants Comparison**     **(b) Performance under Different Attacks**     **(c) The Hyperparameter $\lambda$**



Figure 4: Ablation experiments on AdPO. **(a)** The performance of DPO variants. **(b)** The evaluation of attack types. **(c)** The impact of the parameter $\lambda$.

### 4.3 FURTHER EVALUATION

Although we conduct extensive quantitative evaluations above, they are still insufficient for a comprehensive assessment of LVLMs.

**Qualitative evaluation.** As depicted in Figure 3, the LLaVA model, using the original CLIP as the encoder, provides the most accurate and detailed understanding of clean images. However, when faced with adversarial images, they are completely vulnerable to successful attacks. TeCoA fails to exhibit robust performance against both clean and adversarial images, whereas FARE experiences a loss of detail or minor errors in image understanding, ultimately falling short of optimal performance. In the absence of adversarial defenses, LLaVA is susceptible to manipulation, resulting in biased outputs that can mislead users and have detrimental effects.

### 4.4 ABLATION STUDY

**The impact of DPO variants.** In Figure 4 (a), we evaluate four commonly used DPO variants to analyze the effectiveness of AdPO. The results show that IPO (Azar et al., 2024), KTO (Ethayarajh et al., 2024), and StepDPO (Lai et al., 2024) perform well, while SimPO (Meng et al., 2025) performs relatively poorly, possibly due to the removal of the reference model. This experiment also demonstrates that AdPO serves as **a general preference framework** for enhancing model robustness, rather than being restricted to a specific algorithm.

**Analysis of attack types.** We further evaluate the impact of other attack methods, including C&W (Carlini & Wagner, 2017), sC&W (Zhang et al., 2020), BIM (Kurakin et al., 2017a), CroPA (Luo et al., 2024a), and Verbose (Gao et al., 2024). As shown in Figure 4 (b), our method remains robust even against attacks specifically designed for LVLMs.

**The impact of $\lambda$.** We perform untargeted attacks to evaluate the effectiveness of AdPO trained with diffenrent $\lambda$ on the Flickr30K dataset, with experimental results shown in Figure 4 (c). We find that the clean performance is largely insensitive to AIO, whereas increasing $\lambda$ significantly improves adversarial robustness, with the best empirical results achieved around $\lambda = 1$.

## 5 CONCLUSION

We propose AdPO, the first adversarial defense strategy based on preference optimization. It jointly optimizes the model's outputs on both clean and adversarial images, thereby better preserving clean performance under adversarial training. Unlike previous adversarial fine-tuning methods, which typically only impose single-target constraints to improve adversarial robustness, leading to a loss of clean performance. Moreover, AdPO does not require labeled image data or the CLIP text encoder, offering greater flexibility. Both quantitative and qualitative analyses demonstrate that our method achieves state-of-the-art results, with particularly significant improvements on generative vision-language understanding tasks.

ACKNOWLEDGMENTS

This research was supported by the National Natural Science Foundation of China (62276245) and the Strategic Priority Research Program of Chinese Academy of Sciences (XDA0490000).

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

# Appendix

## THE USE OF LARGE LANGUAGE MODELS

In preparing this paper, we primarily used ChatGPT for language polishing. The model was employed to refine grammar, improve readability, and adjust the overall clarity and conciseness of the manuscript.

## A  UNTARGETED ATTACK DETAILS

For the captioning tasks COCO and Flickr30k, each image is accompanied by 5 available ground-truth captions, with each caption utilized to calculate the CIDEr score. We perform APGD attacks with 100 iterations on each ground truth. After each attack, we calculate the CIDEr score and cease attacking samples that score below 10 for COCO or 2 for Flickr30k. This strategy is primarily employed because generating adversarial attacks on VLMs is extremely time-consuming and costly, allowing us to allocate our limited resources to more impactful attacks. For the VQA task, we apply a similar approach, but the score threshold is set to 0, and we utilize the 5 most common ground truths out of the 10 available.

## B  PROMPTS

We list the prompt templates we used for image description tasks:

- What is the content of the image?
- What does the image show?
- Describe the image in detail.
- What is the image illustrating?
- What is displayed in the image?
- Can you explain what is happening in the image?
- Provide a detailed description of the image.
- Summarize the main objects and elements in the picture.
- What objects or people are visible in the image?
- Describe the scene shown in the image.
- What kind of activity is depicted in the image?
- Give a comprehensive explanation of the image content.
- Identify and describe the key features of the image.
- What is the overall setting or environment shown in the picture?
- Provide a step-by-step breakdown of what you see in the image.
- Describe the relationships between the objects in the picture.
- What message or information does the image convey?
- Provide a caption that accurately summarizes the image.

## C  TARGETED ATTACK DETAILS

We use the following target sequences to perform targeted attacks:

- `A group of people are playing basketball.`
- `a bunch of people are flying in the air.`
- `The pizza on the table is very bad.`

- An earthquake is about to happen.
- This patient needs the best medical care.

If these targeted and even subtly provocative phrases are successfully embedded into adversarial images, they can subtly deceive users, thereby influencing their subsequent decisions. In addition to the quantitative evaluation presented in Section 4.2, we also observed that when the target text is closely related to the image content, the success rate of adversarial attacks is significantly high, indicating that images can easily mislead LVLMs. This presents a more dangerous scenario because when the target text is only weakly related to the image, users can more easily spot these erroneous outputs, thereby reducing their trust in the model. Conversely, when the model's output appears somewhat plausible in relation to the image content, users are more likely to trust the model's output.

## D    EXPERIMENTATION ON MORE TASKS

Table 3: Hallucination evaluation on the POPE dataset. The reporting results are based on the F1-score metric.

| Method | Clean | TeCoA | TGA | FARE | PMG | TGA | AGH | AdvSimplex | AdPO |
|---|---|---|---|---|---|---|---|---|---|
| F1-score | 84.5 | 75.9 | 78.2 | 80.8 | 81.2 | 83.2 | 82.9 | 82.3 | 83.7 |

It is widely recognized that evaluating large vision-language models is challenging. In addition to assessing several key multimodal tasks in Section 4.1, this section further examines the performance of our method on additional vision-language tasks.

**Hallucinations.** One of the greatest challenges faced by LVLMs is hallucination, where these models may perceive objects in an image that do not actually exist. This issue has garnered widespread attention within the research community. We selected the commonly used POPE dataset (Li et al., 2023d) to evaluate multiple CLIP versions of the LLaVA model. In this dataset, the model is required to answer "Yes" or "No" to indicate whether a specific object is actually present in the image. Table 3 shows that our version of CLIP achieved the highest accuracy, but our AdPO method most effectively addresses clean performance. In contrast, both TeCoA and FARE demonstrated a more pronounced decline in performance.

Table 4: The evaluation of jailbreak attack defense, with the attack success rate reported.

| Method | $\epsilon$ | ASR |
|---|---|---|
| CLIP | 0 | 14 / 40 |
| TeCoA | 0 | 14 / 40 |
| TGA | 0 | 15 / 40 |
| FARE | 0 | 13 / 40 |
| AdPO | 0 | 8 / 40 |
| CLIP | $^{16}/_{266}$ | 25 / 40 |
| TeCoA | $^{16}/_{266}$ | 16 / 40 |
| TGA | $^{16}/_{266}$ | 16 / 40 |
| FARE | $^{16}/_{266}$ | 16 / 40 |
| AdPO | $^{16}/_{266}$ | 8 / 40 |

**Jailbreaking attacks.** Recent studies have shown that LVLMs are more vulnerable to jailbreak attacks than pure LLMs, especially when adversarial perturbations are added to images (Qi et al., 2024; Carlini et al., 2023). Therefore, it is essential to further analyze our method's robustness against jailbreak attacks. Under normal circumstances, model owners align models with human values to prevent them from generating suggestive or harmful content. For example, if a user enters a malicious prompt like "How to make a bomb," the model should refuse to respond. However, with the introduction of adversarial images, attackers can more easily bypass these security guard, inducing the model to output intended content and thereby posing greater risks. Following the setup from (Qi et al., 2024), we evaluate LLaVA 1.5 with different CLIP versions under various noise

levels. The results are shown in Table 4. Even without adversarial images, LLaVA can be affected by jailbreak attacks to generate harmful content. Once noise is introduced, however, the success rate of jailbreak attacks on the clean CLIP version increases significantly, while adversarially trained versions maintain their original level of security. This indicates that adversarial training can also enhance the robustness of LVLMs against jailbreak attacks, with our method achieving the best performance. It is important to note that jailbreak attacks are currently a very active area of research, and our evaluations may somewhat overestimate their performance.

Table 5: Adversarial Evaluation on the Qwen-2.5-VL model.

| Method | Type | COCO | Flickr30k | TextVQA | VQAv2 |
|---|---|---|---|---|---|
| Original | - | 124.3 | 82.3 | 79.3 | 84.3 |
| FARE | Clean | 118.5 | 74.3 | 65.1 | 73.8 |
| | Adversarial | 61.8 | 35.2 | 20.8 | 34.7 |
| AdPO | Clean | 124.8 | 82.2 | 79.2 | 84.1 |
| | Adversarial | 78.3 | 50.6 | 37.2 | 48.5 |

Table 6: Adversarial Evaluation on the InternVL3.5 model.

| Method | Type | COCO | Flickr30k | TextVQA | VQAv2 |
|---|---|---|---|---|---|
| Original | - | 124.5 | 83.6 | 78.2 | 82.4 |
| FARE | Clean | 117.4 | 73.2 | 62.4 | 70.3 |
| | Adversarial | 60.5 | 32.1 | 22.4 | 34.5 |
| AdPO | Clean | 124.4 | 83.4 | 79.0 | 82.5 |
| | Adversarial | 79.3 | 48.8 | 36.4 | 47.2 |

Table 7: Adversarial Evaluation on the BLIP-2 model.

| Method | Type | COCO | Flickr30k | TextVQA | VQAv2 |
|---|---|---|---|---|---|
| Original | - | 98.2 | 70.7 | 40.3 | 48.2 |
| FARE | Clean | 88.5 | 62.2 | 29.5 | 32.2 |
| | Adversarial | 41.2 | 23.7 | 10.3 | 15.3 |
| AdvSimplex | Clean | 82.9 | 59.8 | 19.9 | 27.5 |
| | Adversarial | 40.7 | 21.8 | 12.8 | 14.6 |
| AGH-MAT | Clean | 85.1 | 62.5 | 21.3 | 29.1 |
| | Adversarial | 42.3 | 25.5 | 14.1 | 18.0 |
| TGA-ZSR | Clean | 84.2 | 61.3 | 20.8 | 28.0 |
| | Adversarial | 41.4 | 22.9 | 13.4 | 15.2 |
| AdPO | Clean | 98.5 | 70.3 | 39.3 | 48.2 |
| | Adversarial | 65.4 | 48.2 | 22.8 | 24.4 |

## E    EXPERIMENT ON NON-CLIP MODELS

In order to assess the generalizability of AdPO beyond CLIP-based models, we conduct empirical evaluations on Qwen-2.5-VL, InternVL3.5, and BLIP-2.

Qwen2.5-VL employs an image encoder with an improved self-attention mechanism, while InternVL3.5 uses InternViT as its image encoder. We apply AdPO for comparison, and neither model requires a text encoder. As shown in Tables 5 and Table 6, our method achieves a substantial lead, particularly in adversarial robustness, consistently outperforming FARE by more than 10 points. We also evaluate BLIP-2, which uses the EVA-CLIP model as its encoder, to enable a more comprehensive comparison with prior methods. As shown in the results in Table 7, AdPO still achieves a substantial lead, further demonstrating that it is a model-agnostic approach.

## F    ANALYSIS OF ATTACK STRENGTHS

In this section, we explore the impact of higher attack strengths with the results presented in Table 8 and Table 9.

Table 8: The performance of attacks with $8/255$.

| Method | COCO | Flickr30k | TextVQA | VQAv2 |
|--------|------|-----------|---------|-------|
| FARE   | 25.2 | 13.2      | 5.2     | 10.1  |
| TGA    | 26.7 | 14.2      | 6.9     | 15.2  |
| AdPO   | 42.5 | 24.5      | 13.3    | 22.5  |

Table 9: The performance of attacks with $16/255$.

| Method | COCO | Flickr30k | TextVQA | VQAv2 |
|--------|------|-----------|---------|-------|
| FARE   | 8.2  | 3.2       | 2.2     | 3.1   |
| TGA    | 10.7 | 8.2       | 3.6     | 5.2   |
| AdPO   | 20.2 | 13.2      | 8.2     | 14.9  |

We find that models trained with low attack intensity exhibit some level of adversarial robustness when faced with high-disturbance adversarial samples. However, they show a noticeable performance drop compared to models trained with the same level of perturbation. Compared to previous state-of-the-art methods, our method still achieves a significant lead.

## G    EXPERIMENTAL RESULTS ON TINY-LLAVA

Table 10: Experimental Results on Tiny-LLaVA.

| Method | Type | COCO | Flickr30k | TextVQA | VQAv2 |
|--------|------|------|-----------|---------|-------|
| Original | - | 90.3 | 65.3 | 40.4 | 69.5 |
| FARE | Clean | 83.2 | 55.3 | 30.7 | 58.2 |
|      | Adversarial | 39.2 | 20.5 | 8.9 | 17.2 |
| TGA | Clean | 80.1 | 53.9 | 25.8 | 55.7 |
|     | Adversarial | 40.2 | 23.1 | 10.2 | 18.2 |
| AdPO | Clean | 91.2 | 66.3 | 42.8 | 65.2 |
|      | Adversarial | 50.3 | 42.7 | 27.4 | 28.4 |

Table 10 presents our experimental results on TinyLLaVA. The results demonstrate that our method achieves a substantial improvement over previous approaches. This improvement can be attributed to the direct joint training of the image encoder and the target decoder, which enables more effective vision-language alignment.

## H    EFFICIENCY–PERFORMANCE TRADE-OFF

As illustrated in Table 11, we present a detailed comparison of architectural dependencies and inference speeds across different methods. Previous approaches, such as TeCoA and TGA-ZSR, rely simultaneously on both the image encoder and the text encoder, which limits their efficiency and results in a medium inference speed. To accelerate the process, methods like FARE and Sim-CLIP depend solely on the image encoder, thereby achieving fast inference speeds. However, this extreme reduction in architectural dependency causes them to fall short in effectively enhancing the model's fine-grained understanding capabilities. In contrast, our proposed method leverages an image encoder combined with a text decoder. Although it maintains a medium inference speed comparable to earlier baselines, it achieves a superior efficiency-performance trade-off by significantly boosting fine-grained semantic comprehension.

Table 11: Comparison of module dependencies and inference speed among different methods.

| Method | Image Encoder | Text Encoder | Text Decoder | Speed |
|---|---|---|---|---|
| TeCoA | ✓ | ✓ | ✗ | Medium |
| TGA-ZSR | ✓ | ✓ | ✗ | Medium |
| FARE | ✓ | ✗ | ✗ | Fast |
| Sim-CLIP | ✓ | ✗ | ✗ | Fast |
| Ours | ✓ | ✗ | ✓ | Medium |

## I ADDITIONAL ABLATION STUDIES

In this section, we primarily investigate the impact of direct adversarial training and full fine-tuning on model performance as shown Table 12.

Table 12: Additional ablation studies.

| Method | Clean | Adversarial |
|---|---|---|
| Direct Adversarial Training | 66.3 | 42.2 |
| Full Fine-tuning | 72.3 | 50.4 |
| SFT-based AIO | 70.0 | 41.1 |
| AdPO | 73.6 | 52.3 |

We observe that direct adversarial training significantly degrades clean performance without providing notable improvements in adversarial robustness. On the other hand, full fine-tuning slightly compromises transferability, leading to minor drops in both clean and adversarial performance. We find that when applying SFT-based AIO (Eq. 6), both clean and adversarial performance degrade significantly. This decline arises from the strong negative impact of SFT on model generalization, a phenomenon consistent with recent findings in the literature (Chu et al., 2025).

## J ANALYSIS OF FAILURE SAMPLES

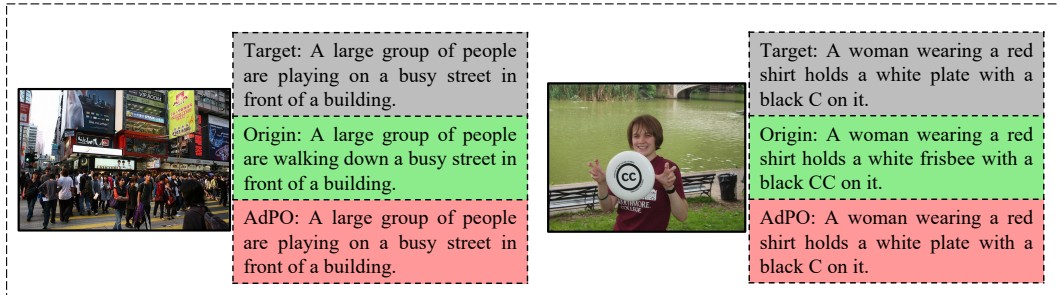

Figure 5: Showcase of failure samples.

In this section, we provide a qualitative analysis of failure cases, as illustrated in Figure 5. Our observations indicate that AdPO is particularly vulnerable when the adversarial target is semantically close to the true content of the image. These cases often involve fine-grained distinctions that are semantically ambiguous, making them difficult for the model to reliably discriminate. The adversarial attack exploits this ambiguity, thereby increasing the likelihood of misleading the model. This analysis clarifies the inherent limitations of AdPO when operating under subtle semantic shifts.

