# OpenReview forum: "AdPO: Enhancing the Adversarial Robustness of Large Vision-Language Models with Preference Optimization"
_ICLR.cc/2026/Conference — ICLR 2026 Poster_

### Official Review · Reviewer_WQTL · 2025-10-26

**Soundness:** 3
**Presentation:** 3
**Contribution:** 2
**Rating:** 6
**Confidence:** 2

**Summary:**

This paper reframes robustness optimization for vision–language models as a preference optimization problem. The authors propose Preferred Image Optimization (PIO) to encourage normal outputs on clean inputs, and Adversarial Image Optimization (AIO) to reject potentially misleading outputs on adversarial examples. Extensive experiments demonstrate the effectiveness of the approach.

Regarding novelty, I am not an expert in preference optimization and am unfamiliar with the detailed trajectory of this line of work, so my assessment of novelty is tentative and I will defer to other reviews.

**Strengths:**

1. The method is simple and effective, with empirical validation across a broad set of experiments.

2. Framing VLM robustness as a preference optimization problem is an interesting perspective.

**Weaknesses:**

1. Motivation is insufficient. On line 90 the authors state: “we explore fine-tuning the image encoder of a smaller LVLM and subsequently transferring it to a larger LVLM model.” While reasonable, I do not view this as a contribution by itself. Cost reduction by using a smaller LVLM is expected; the paper does not explain why this transfer should work or provide guarantees.

2. Typos. For example, in the caption of Figure 3, TeCoA is misspelled as CoTeA.

3. Figure readability. The axes in the three plots of Figure 4 are not clearly visible.

**Questions:**

1. What happens if the encoder is changed? Please provide an ablation on the image encoder choice and discuss sensitivity to encoder architecture/initialization.

---

> ### Author Response · Authors · 2025-11-21
>
> We thank you for your valuable feedback and for recognizing our method as "simple and effective" and an "interesting perspective." We address your specific concerns below:
>
> # Clarification on the contribution and the transfer strategy.
>
> We thank the reviewer for raising this important question regarding the transfer strategy. We would like to take this opportunity to clarify a potential misunderstanding about our contribution.
>
> - Core Contribution: Our paper's core contribution is the AdPO framework itself—the novel reframing of adversarial robustness as a preference optimization problem.
> - Transfer as an Efficiency Strategy: The 'transfer from a smaller LVLM' strategy mentioned by the reviewer is not our primary contribution. Rather, it is an efficient, practical validation we proposed to mitigate the high computational costs associated with training MLLMs. As we explicitly stated in **our contribution summary on Line 108**, this transfer strategy is presented as an **empirical validation** rather than a standalone methodological contribution.
>   > We **validate** the feasibility of conducting adversarial training on smaller LVLMs......
>   >
>
> The reviewer correctly asks why this transfer strategy works. The rationale is based on the fundamental architecture of LVLMs, which consist of a vision encoder and a language model.
> The encoder's role is to extract visual features, which are then aligned to the LLM's semantic space to generate text.
> This paper focuses on the adversarial robustness of LVLMs on the visual side, whose adversarial vulnerability primarily stems from the vision encoder. Perturbations first corrupt the visual features, and these 'toxic' features are then fed to the LLM, causing erroneous outputs.
> Therefore, our approach targets this vulnerability at its source. By enhancing the robustness of only the vision encoder, AdPO effectively 'purifies' the visual features before they ever reach the LLM. This enhanced robustness in the visual-feature space is then naturally conducted to the entire MLLM's output. Because this robustness is embedded in the encoder's weights, the trained encoder can serve as a robust, 'plug-and-play' module for larger LVLMs that share the same architecture.
>
> # Regarding typos and figure readability.
>
> We thank the reviewer for the helpful feedback. All noted typos have been corrected, and Figure 4 has been updated with larger fonts and clearer labels to improve readability in the revised manuscript.
>
> # On Encoder Choice and Generalization.
>
> This is a crucial point regarding our method's generalization. We are happy to confirm that we already included these experiments in **Appendix Section E** of our submission.
> In the appendix, we evaluated AdPO on several other models with different architectures, such as Qwen2.5-VL, InternVL3.5, and BLIP-2, which use a variety of image encoders. The experimental results clearly demonstrate our method's strong generalization.
> Overall, our approach successfully improves the adversarial robustness for both of the mainstream LVLM visual architectures: the direct projection style (represented by LLaVA) and the query-transformer-based style (represented by BLIP-2).
>
> We thank you again for your constructive feedback. We hope these clarifications have fully addressed your concerns.

---

> ### Comment · Area_Chair_JMWc · 2025-11-25
>
> Dear Reviewer WQTL,
>
> The authors have responded to your reviews. Please review and provide your feedback and responses.
>
> Best,
>
> Your AC

---

### Official Review · Reviewer_4Lsf · 2025-10-28

**Soundness:** 4
**Presentation:** 3
**Contribution:** 4
**Rating:** 8
**Confidence:** 4

**Summary:**

This paper proposes AdPO, a novel adversarial defense strategy for Large Vision-Language Models (LVLMs) based on preference optimization. The key insight is to reframe adversarial training as a preference learning problem, encouraging the model to favor correct outputs for clean inputs while rejecting misleading ones for adversarially perturbed inputs. The method combines Preferred Image Optimization (PIO) and Adversarial Image Optimization (AIO), the former maintaining clean accuracy and the latter enhancing robustness through dynamic fine-tuning.

Experiments across multiple LVLM benchmarks demonstrate that AdPO achieves state-of-the-art robustness with minimal degradation on clean performance. The paper also shows efficient transferability by training only the image encoder of small models and transferring it to larger LVLMs.

**Strengths:**

1. The study of adversarial training for LVLMs is timely and relevant. While many recent works have focused on CLIP-style vision-language models, the increasing real-world deployment of LVLMs (e.g., Gemini) makes it necessary to explore adversarial training specifically tailored for such models.
2. Figure 2 effectively helps readers quickly grasp the core idea of the AdPO method.
3. I particularly appreciate AdPO’s adversarial modeling approach based on preference optimization. It effectively optimizes two complementary objectives: increasing the probability of generating correct outputs for clean inputs while reducing the likelihood of producing malicious outputs for adversarial inputs. Moreover, it explicitly optimizes the probability of generating correct outputs under adversarial perturbations. This innovative training paradigm enables the model to achieve both excellent clean accuracy and strong adversarial robustness.
4. Extensive experiments across multiple benchmarks and models demonstrate the effectiveness and generality of the proposed method.

**Weaknesses:**

1. Typo error — the citation for CroPA in Line 467 is missing parentheses.
2. The experiments primarily evaluate results under low attack intensities (2/255, 4/255). Evaluating higher attack strengths (8/255, 16/255) would make the assessment more comprehensive.
3. PIO is described as a relative preference, yet $y_l$ is not necessarily incorrect in every generation, which could potentially affect the results.
4. The subheadings in Figure 4 appear to be a bit too small.

**Questions:**

Q1: If I understand correctly, in the main experiments neither the baseline methods nor AdPO have seen the subsequent language models during training. I would like to get confirmation from the authors that the comparison is entirely fair.

For other questions, please see Weaknesses

Overall, I believe this paper is a solid, well-executed work with clear motivation and convincing results.

---

> ### Author Response · Authors · 2025-11-21
>
> We are grateful to you for your incredibly positive and encouraging review. We are delighted that you found our work's contribution and soundness "excellent," the "training paradigm innovative," and the topic "timely and relevant."
> To respond to your concerns in detail, we offer the following clarifications:
>
> # On experiments with higher attack strengths (8/255, 16/255).
>
> We completely agree that this is crucial for a comprehensive assessment. In fact, we did include these experiments in our original submission's Appendix. We would kindly refer the reviewer to Tables 4, 8, and 9 in the Appendix, which present the results for these higher attack intensities. These results demonstrate that our method, AdPO, maintains strong robustness even under these more severe perturbations.
>
> # On the PIO negative sample potentially being correct.
>
> This is a very sharp and insightful observation. We acknowledge this possibility, and our method is designed to be robust to it, as briefly discussed in **Line 273** of the submission. In the worst-case scenario where the sample $y_l$ happens to be identical to $y_w$, the preference margin becomes zero, and that specific sample simply produces no effective gradient. This allows the process to "fail safe" for that single instance without harming the overall training.
>
> # Regarding minor presentation issues.
>
> We thank the reviewer for pointing these out. The citation for CroPA in Line 467 has been corrected, and Figure 4 has been updated with larger subheadings and clearer axis labels to improve readability in the revised version.
>
> # Confirmation of fair comparison.
>
> Yes. We would like to absolutely confirm that your understanding is correct. The comparison is entirely fair. All baseline methods were trained under the exact same conditions, and none of them (including AdPO) had access to the subsequent, larger language models that were used during the final evaluation.
>
> We thank you once again for your support and helpful suggestions, which have helped us further polish the paper.

---

> > ### Comment · Reviewer_4Lsf · 2025-11-25
> >
> > Thank you for the authors’ detailed and thoughtful responses. The rebuttal has resolved my concerns. The work is solid and the experiments are comprehensive. I maintain my score and support accepting this paper.

---

> > > ### Author Response · Authors · 2025-11-25
> > >
> > > Dear Reviewer 4Lsf,
> > >
> > > Thank you for your positive feedback. We are glad that our responses have addressed your questions. We greatly appreciate your time, your constructive comments, and your strong support for our paper.
> > >
> > > Best regards,
> > >
> > > Authors

---

> ### Comment · Area_Chair_JMWc · 2025-11-25
>
> Dear Reviewer 4Lsf,
>
> The authors have responded to your reviews. Please review and provide your feedback and responses.
>
> Best,
>
> Your AC

---

### Official Review · Reviewer_5ypa · 2025-10-29

**Soundness:** 3
**Presentation:** 3
**Contribution:** 3
**Rating:** 6
**Confidence:** 4

**Summary:**

This paper introduces AdPO, a novel adversarial training framework built upon preference optimization (DPO). Specifically, the authors propose Preferred Image Optimization (PIO) and Adversarial Image Optimization (AIO) to simultaneously enhance clean-image performance and adversarial robustness. The formulation of adversarial robustness through preference optimization is insightful, and the experiments demonstrate strong performance across multiple datasets. Furthermore, the method exhibits excellent transferability to larger LVLMs, achieving state-of-the-art results.

**Strengths:**

1. The formulation of adversarial robustness through preference optimization is insightful.
2. The performance of the proposed method demonstrate the effectiveness.
3. The paper is easy to follow and golocally presented.

**Weaknesses:**

1. Compared to previous methods such as TeCoA, AdPO requires substantially more computation. In particular, it needs to generate full responses for both the clean and adversarial images, which makes the overall training cost extremely high. It would be desirable if similar performance could be achieved with offline response generation.
2. In Equation (7) (line 299), it seems that y_l should be y_w instead.

**Questions:**

In Equation 5, what is y_w and y_l? Is y_w the response from clean image and y_l is the response from adversarial image?

---

> ### Author Response · Authors · 2025-11-21
>
> We sincerely thank you for your feedback and for recognizing the insightful formulation of our work, its demonstrated effectiveness, and its clear presentation.
>
> # On computational cost and offline generation.
>
> We appreciate the reviewer’s thoughtful comments regarding computational overhead. While AdPO is more computationally intensive  than earlier methods such as TeCoA, we believe this additional cost is both justified and effectively mitigated in practice.
>
> - Justification for Cost:
>   - Broader Scope and Architectural Generality: Methods like TeCoA were primarily designed for vision-language understanding models (e.g., CLIP). Our AdPO is engineered for the complex demands of modern generative LVLMs (e.g., LLaVA). Crucially, our framework is not limited to a single architecture. As we demonstrate extensively in Appendix (e.g., Section E), AdPO shows strong generalization and effectiveness across a diverse range of LVLM families, including Qwen2.5-VL, InternVL3.5, and BLIP-2. This proves our method's wide applicability to different visual backbones and architectural styles (e.g., direct projection vs. Q-Former based).
>   - Superior Performance-Robustness Trade-off: This cost corresponds to a very significant performance improvement. AdPO achieves state-of-the-art robustness while, critically, maintaining high clean-image performance. This ability to solve the robustness-accuracy trade-off is a key challenge that previous methods struggled to achieve.
> - Efficiency Strategy: To address the overall training burden, we proposed the highly effective strategy of training only the image encoder on a smaller MLLM and transferring the robust encoder to larger models. As our experiments demonstrate, this makes our approach practical and computationally efficient in aggregate.
> - Regarding Offline Generation: This is an insightful suggestion. However, adversarial examples generated "offline" from a stale model would be of significantly lower quality as negative samples, as they would not target the vulnerabilities of the model as it is being trained. This would lead to sub-optimal robustness. How to create high-quality, effective offline training data for LVLM robustness is a valuable and open research question.
>
> # Regarding the typo in Equation (7).
>
> We sincerely thank you for spotting this error. You are absolutely correct that $y_l$ should be $y_w$ in Equation (7) (Line 299). This typo has been corrected in the revised manuscript.
>
> # Regarding the definitions of $y_w$ and $y_l$ in equation 5.
>
> Thank you for the opportunity to clarify this. Your intuition is exactly correct, where $y_w$ is the preferred response (to the clean image) and $y_l$ is the dispreferred response (generated from the adversarial image).
>
> We thank you again for your valuable and expert feedback, which has helped us improve the paper.

---

> ### Comment · Area_Chair_JMWc · 2025-11-25
>
> Dear Reviewer 5ypa,
>
> The authors have responded to your reviews. Please review and provide your feedback and responses.
>
> Best,
>
> Your AC

---

> > ### Comment · Reviewer_5ypa · 2025-11-25
> >
> > Thanks for the rebuttal. Can you provide a table that summarizes the efficiency–performance trade-off?

---

> > > ### Author Response · Authors · 2025-11-25
> > >
> > > Thank you for your response. We would like to clarify that we have already included an efficiency comparison in Table 11 of the original submission. The results show that our method incurs only a slight overhead compared to the baseline.

---

### Official Review · Reviewer_E9TV · 2025-10-30

**Soundness:** 2
**Presentation:** 3
**Contribution:** 3
**Rating:** 6
**Confidence:** 2

**Summary:**

This paper proposes AdPO, a preference-optimization–based adversarial defense for large vision-language models (LVLMs). By reframing adversarial training as a preference learning problem, AdPO enhances robustness while preserving clean performance. The method introduces Preferred Image Optimization (PIO) and Adversarial Image Optimization (AIO), applied only to the image encoder for efficiency and transferability. Experiments on multiple LVLM benchmarks show that AdPO achieves state-of-the-art robustness without significant degradation on clean data.

**Strengths:**

1. The paper is well written and clearly structured, making the methodology and experimental results easy to follow.

2. The motivation is sound and well justified, addressing the important problem of improving adversarial robustness in large vision-language models.

3. The work presents a creative combination of existing ideas, integrating preference optimization with adversarial training for LVLMs.

4. The proposed defense method demonstrates strong and consistent performance, significantly improving adversarial robustness while maintaining clean accuracy.

5. The experimental evaluation is comprehensive and convincing, covering multiple open-source LVLM architectures, diverse benchmarks, and an analysis of various DPO variants to validate the generality and robustness of the approach.

**Weaknesses:**

1. The paper does not analyze potential failure cases of AdPO. It would be helpful to evaluate its robustness under adaptive attacks (where the attacker knows the defense) and transfer-based attacks generated from unseen models, to better understand the method’s practical limitations.

2. While the empirical performance of AdPO is impressive, the paper lacks a theoretical discussion connecting its objective to adversarial risk minimization. The approach largely builds upon the known DPO framework, and providing a clearer theoretical justification would strengthen the work’s originality and rigor.

3. (Minor) There are several typos in the submission. For example, in the caption of Figure 3, it should be TeCoA; in line 471, it should be “Flickr30K”; and there are similar minor errors around line 376. Careful proofreading is recommended before the camera-ready version.

**Questions:**

1. How would AdPO perform under transfer-based (black-box) attacks generated from unseen models?

2. Could the authors provide examples or qualitative analyses of failure cases, showing when AdPO fails and what patterns those failures share? This would help readers understand the limitations and potential future improvements.

3. Can the authors clarify whether AdPO’s optimization objective can be interpreted as a form of adversarial risk minimization? If so, what assumptions or approximations are required for this equivalence?

4. Is there a theoretical intuition or derivation showing how preference optimization contributes to robustness beyond empirical observations?

---

> ### Author Response · Authors · 2025-11-21
>
> We sincerely thank you for your constructive feedback and for recognizing the key strengths of our work, including its clear presentation, sound motivation, the creative integration of preference optimization, and the strong, consistent performance across comprehensive experiments. We have revised our paper based on your suggestions and address the specific concerns below.
>
> # Regarding the analysis of failure cases.
>
> This is an excellent suggestion. We have added a qualitative analysis of failure cases in the Appendix (new Figure 5).
> Our analysis reveals that AdPO is more prone to failure when the attack target is semantically close to the image's true content.
> We hypothesize this is because such fine-grained errors are "specious", and they are semantically ambiguous even for the model itself. The adversarial attack effectively exploits this ambiguity to mislead the model.
>
> We thank your for this suggestion, as it has helped us to more clearly define the limitations of our method.
>
> # Regarding transfer-based (black-box) attacks.
>
> We thank you for this valuable question, and we would like to clarify that in the field of Multimodal Large Models (MLLMs), transfer-based black-box attacks are currently at a nascent stage and demonstrate significantly lower efficacy than white-box attacks. Given that our paper's primary focus is on adversarial training (a defense method), we believe that evaluating our method against the strongest and most direct threat, white-box attacks, is the most reasonable and rigorous approach.
>
> Nevertheless, to address your concern, we have conducted a new experiment on black-box attacks. We performed a transfer attack from MiniGPT-4 (as the source attack model) to LLaVA-1.5 (as the victim model) on the COCO dataset.
>
> | Model     | Original Performance (Clean) | Black-box Attack Performance |
> | --------- | ---------------------------- | ---------------------------- |
> | LLaVA-1.5 | 115.3                         | 113.3                         |
>
> As the results show, the impact of this black-box attack on the model's performance is minimal. This aligns with our understanding that current transfer-based attacks for MLLMs have limited effectiveness, and designing potent black-box attacks for MLLMs remains a significant long-term research challenge in the field.
>
> # Can AdPO’s objective be interpreted as adversarial risk minimization?
>
> We respectfully clarify that to some extent, our objective can be interpreted as a form of adversarial risk minimization, but it operates from a pairwise preference perspective rather than a traditional point-wise loss perspective.
> Our formulation assumes that $y_l$ corresponds to an incorrect output; however, as discussed in Line 270, our implementation is not strongly dependent on this assumption and demonstrates notable robustness.
> Overall, this work is primarily empirical in nature, aiming to validate the effectiveness of our approach through comprehensive experiments, while we leave a deeper theoretical analysis for future work.
>
> # Is there a theoretical intuition for how preference optimization contributes to robustness?
>
> Our theoretical intuition is primarily drawn from a multi-objective optimization perspective.
> The core challenge of adversarial training (AT) lies in balancing two often-conflicting objectives: Enhancing robustness on adversarial inputs and Maintaining performance on clean, benign inputs.
> This challenge directly maps to the concept of 'winner' (preferred) and 'loser' (dispreferred) sample pairs in preference optimization. This conceptual alignment inspired us to apply a preference-based learning framework to adversarial training.
>
> # Regarding minor typos.
>
> We are grateful to you for your careful proofreading. We have corrected all the mentioned typos in the newly submitted version and will conduct a comprehensive check in the final camera-ready version.
>
> We once again thank you for your valuable comments, which have significantly helped us improve the quality of our paper. We hope our responses and revisions have fully addressed your concerns.

---

> ### Comment · Area_Chair_JMWc · 2025-11-25
>
> Dear Reviewer E9TV,
>
> The authors have responded to your reviews. Please review and provide your feedback and responses.
>
> Best,
>
> Your AC

---

> ### Comment · Reviewer_E9TV · 2025-11-25
>
> Thank you for the detailed rebuttal. I have no further comments on the experimental part; it is solid and comprehensive. However, if the authors would like to further discuss the theoretical justification of AdPO, I would be happy to engage. I maintain my score 6.

---

> > ### Author Response · Authors · 2025-11-25
> >
> > Dear Reviewer E9TV,
> >
> > We sincerely thank you for the time and effort invested in assessing our work. Your constructive feedback has been greatly helpful in improving the overall quality of the paper.
> > Regarding the theoretical justification, we value your insight and agree it is a promising direction for future research.
> >
> > Thank you again for your constructive feedback throughout the review process.
> >
> > Best regards,
> >
> > Authors

---

### Meta-Review · Area_Chair_kLUv · 2026-01-08

**Summary:**

Reviewers agree AdPO reframes LVLM adversarial training as preference optimization (PIO/AIO) and, by updating only the vision encoder, achieves strong robustness–clean trade-offs with broad empirical validation and transfer to larger LVLMs; remaining concerns are mainly about training cost/efficiency reporting and limited theoretical grounding rather than empirical weakness.

**Reviewer Concerns:**

The rebuttal substantially addresses practical questions by adding failure-case analysis, reporting (limited) black-box transfer results, clarifying the preference formulation/typo fixes, pointing to existing higher-intensity attacks and multi-architecture evaluations, and explaining why encoder-only transfer is plausible; the main outstanding issue is still a lightweight but clearer theoretical connection to adversarial risk minimization/calibration (and a crisper efficiency–performance summary), though these do not seem to undermine the contribution.

**Reviewer Scores:**

Under full discussion, I expect Reviewer 4Lsf stays at 8 (explicitly reaffirmed), Reviewer E9TV stays at 6 (explicitly maintained, theory still open), Reviewer 5ypa stays at 6 (typo resolved; efficiency table request partially answered by pointing to Table 11).

WQTL did not reply, but the authors’ rebuttal appears to address their key points: they clarify that “small-to-large transfer” is an efficiency strategy (not the core contribution), fix typos/figures, and point to appendix experiments across different LVLMs/encoders to support robustness under encoder changes. Given this, I expect WQTL would likely keep the score at 6 (or at most slightly increase)

---

### Decision · Program_Chairs · 2026-01-26

Accept (Poster)